# Trustworthy Environmental Monitoring Using Hardware-Assisted Security Mechanisms

**DOI:** 10.3390/s24144720

**Published:** 2024-07-20

**Authors:** Laurent Segers, Borna Talebi, Bruno da Silva, Abdellah Touhafi, An Braeken

**Affiliations:** 1Department of Engineering Technology (INDI), Vrije Universiteit Brussel (VUB), 1050 Brussels, Belgium; laurent.segers@vub.be (L.S.); abdellah.touhafi@vub.be (A.T.); 2Department of Electronics and Informatics (ETRO), Vrije Universiteit Brussel (VUB), 1050 Brussels, Belgium; bruno.da.silva@vub.be

**Keywords:** Secure IoT, TrustZone, Secure IoT communication, key agreement protocol, low-power Secure IoT

## Abstract

Environmental monitoring is essential for safeguarding the health of our planet and protecting human health and well-being. Without trust, the effectiveness of environmental monitoring and the ability to address environmental challenges are significantly compromised. In this paper, we present a sensor platform capable of performing authenticated and trustworthy measurements, together with a lightweight security protocol for sending the data from the sensor to a central server anonymously. Besides presenting a new and very efficient symmetric-key-based protocol, we also demonstrate on real hardware how existing embedded security modules can be utilized for this purpose. We provide an in-depth evaluation of the performance and a detailed security analysis.

## 1. Introduction

The Internet of Things (IoT) has undergone tremendous growth during the last decade [1]. The utilization of IoT for environmental monitoring enables remote monitoring at scale, facilitating cost-effective solutions and empowering data-driven decision-making processes for environmental management [2]. Applications include home automation, wearable technologies, medical applications, manufacturing, predictive maintenance, and environmental monitoring.

Extensive research exists on either developing cost-optimal sensors [3] or setting up a prototype sensor network [4,5]. Most deployed IoT devices sense the environment in which they are deployed and transmit the measured values to a remote server. Security and, in particular, authentication are in most of the related works not included. In many cases, these IoT devices interact with private and sensitive data and should, therefore, offer a certain level of security [6]. Modern consumer-oriented devices such as smartphones, computers, and servers offer data encryption and hashing mechanisms to guarantee the privacy and authenticity of the transferred and stored data. IoT devices, however, are typically constrained in their computation capabilities and do not offer advanced security possibilities. Several solutions have been proposed with the aim of enhancing the security of low-power embedded devices. One of the proposed solutions is based on code isolation between a trusted secure and a user application running on the same chip, which might be presumably insecure. This mechanism is proposed in ARM-Cortex M23 and ARM-Cortex M33 microcontroller-based families via TrustZone [7,8]. Many microcontrollers also offer a variety of cryptographic accelerators implemented in hardware allowing for the local encryption and hashing of data. Aside from security features included in microcontrollers, dedicated crypto-authentication modules augment the security of embedded devices when directly connected to microcontrollers. Other features of current technologies include tamper protection, true random number generators, secure storage solutions, secure boot, root of trust, etc. In recent research, the emergence of physical unclonable function (PUF) has been exploited [9]. PUF technologies exploit the uniqueness of the silicon used in every manufactured integrated circuit. This uniqueness can be exploited via a random source entropy to generate true random numbers via a dedicated portion of SRAM memory in which a given input value produces a repeatable but unique output value. These PUFs allow for security enhancement by contributing to the authentication and key management process [10,11].

In this paper, we investigate the security capabilities offered by modern embedded devices that can be utilized in IoT platforms for environmental monitoring applications. The primary aim is to propose and develop more secure embedded solutions that can be implemented on today’s available technologies. AES-based encryption embedded accelerators will be utilized to encrypt and decrypt the transmitted messages. AES relies on symmetric secret keys known to both the embedded device and the server. These keys, although securely stored on both devices, need to be renewed at certain time intervals to enhance data privacy. This is not an evident task to perform in an effective manner, taking into account the protection of a wide range of attacks, like insider attacks, synchronization attacks, replay attacks, etc. [12]. To this end, we also propose a new lightweight symmetric-key agreement security protocol between the embedded system and the server for symmetric-key renewal.

The paper is structured as follows: Section 2 discusses related work. In Section 3, the preliminaries are provided. In Section 4, the hardware architecture and features of the IoT platform are described. Section 5 deals with a new key agreement protocol for generating symmetric secret keys, and the security analysis of this protocol is presented in Section 6. The performance analysis of the proposed system is discussed in Section 7. Finally, Section 8 concludes the paper.

## 2. Related Work

We split the related work into three parts. First, we delve a bit deeper into other environmental monitoring systems and address how security is handled in those systems. Then, we provide some background on currently available low-power hardware platforms used to implement security features. We also explain the motivation behind our choice of platform. Finally, we discuss the related work on security protocols, in particular client–server protocols, which are the type of protocol present in this setting.

### 2.1. Environmental Monitoring Systems

Currently, larger cities tend to foster the liveability of urban areas. With the advent of many low-cost sensing options and the development of affordable IoT platforms, many opportunities have arisen to monitor the quality of urban environments. Many environmental variables can be sensed, including temperature, relative humidity, acoustic sound pressure level (SPL), perceived light intensity, infrared and ultraviolet light, fine-particle sensing, carbon dioxide levels, etc. While each of the sensed variables on the level of the city is a relevant indicator of the global city evolution, more fine-grained sensing offers local variabilities. Indeed, micro-climatic effects are sometimes observed as many structures tend to accumulate absorbed heat during the daytime while releasing that heat at night time. Air pollution and acoustic pollution have been shown to decrease the enjoyability of certain areas [13].

While sensing the environmental parameters at the local level is an important aspect, the transmission of these values to a remote server also plays an important role. The server then collects the obtained values from all deployed devices, and several tendencies can be drawn. From these tendencies, recommendations for urban (re)arrangements can be derived to enhance the liveability of cities.

Several research and commercial applications can be found in the recent literature. Many include sensing solutions deployed at fixed locations, while some approaches have been implemented by means of moving vehicles. Bobulski et al. [14] developed an air-quality sensing solution designed to be mounted on the roof of automated vehicles. The aim is to measure the concentrations of fine particles (i.e., PM_2.5_ and PM_10_); volatile organic compounds (VOCs); levels of CO, CO_2_, NO, NO_2_, and SO_2_; temperature; and humidity along the trajectory taken by the vehicle. The collected measurements are first transmitted to a local onboard processing device in the vehicle via the Message Queue Telemetry Transport (MQTT) protocol. Once local processing is finalized, the obtained data are transmitted via the Amelia [15] protocol with support for symmetric-key encryption over the existing 4G network to a central server in which post-processing is applied. The results are displayed via heatmaps overlaid with regular street maps, indicating the levels of each of the measured components. A very similar approach is provided by the “Snuffelfiets” device developed in Utrecht [16]. The main goal of “Snuffelfiets” is to measure the levels of fine particles (i.e., PM_1_, PM_2.5_, and PM_10_) and provide cyclists with a safer alternative route in terms of air quality. The sensing device collects data from several onboard sensors and transmits the data in real time to a remote server via the LTE-M network. Thanks to the onboard GPS module, the level of pollutants can be mapped onto a street map. The system is funded by the Rijksinstituut voor Volksgezondheid en Milieu (RIVM) and is being tested by several hundreds of cyclists. There is, however, no information available regarding security features implemented.

Tellez, M. et al. [17] used TelosB nodes in a fixed Zigbee mesh network to monitor temperature and humidity in their setup. The TelosB nodes feature an MSP430 microcontroller, which is attached to a CC2420 IEEE 802.15.4 Zigbee compatible transceiver. A coordinator node acts as a gateway between a remote computer and the Zigbee network, while the devices inside the Zigbee network measure humidity and temperature. The devices inside the Zigbee network use the AES-128 cryptographic module integrated into the CC2420 IEEE 802.15.4 transceiver to encrypt and decrypt the transmitted packets. The coordinator node decrypts these packets before transmitting them to the remote computer. The packets from the computer are encrypted when traveling in the opposite direction. Instead of trying to attack the system via the communication channel, the authors applied a direct physical attack on the motes by successfully brute-forcing the integrated USB bootstrap loader (BSL) of the MSP430. The MSP430-BSL is password-protected and should prevent unauthorized users from reprogramming the application without providing the proper password. However, by brute-forcing this password, one can introduce malicious firmware into the system. The AES key can then be retrieved and reused in malicious firmware, compromising the integrity of the whole sensor network.

Jebari et al. [18] utilized ZigBee to collect gas measurements from nodes inside a poultry farm. Their goal was to monitor the levels of gasses such as ammonia (NH_3_), CO, CO_2_, methane (CH_4_), and H_2_S. These gasses are known to cause excess mortality in poultry farms when present at too high levels. These data are combined with information from nodes outside the farm that use WiFi to transmit the data. The authors designed a “Poultry-Edge-AI-IoT” system, which consists of a heterogeneous platform wherein different types of sensor nodes collect sensory information and transmit it to an AI-enabled processing platform. The sensor nodes inside the barns are equipped with a ZigBee-enabled module, while the nodes placed outside on the roof are equipped with WiFi and Bluetooth transceivers. The combination of nodes inside and outside the buildings allows us to make appropriate decisions with respect to high levels of noxious gasses. While communication is secured via techniques present in both the ZigBee and WiFi/Bluetooth communication schemes, the researchers also made use of RSA encryption algorithms in combination with the SHA256 algorithm in a blockchain manner to make sure no transmitted data are altered.

In [19], multiple encryption algorithms were combined to ensure data integrity. The authors developed an IoT network that measures carbon monoxide, carbon dioxide, temperature, oxygen, and humidity in airports. The sensors use AES 256-bit key encryption in Cipher Block Chaining (CBC) mode. Furthermore, data from the network are sent using HyperText Transfer Protocol Secure (HTTPS) and Secure Sockets Layer (SSL) connections to remote servers as well as SHA-3 to ensure data integrity.

### 2.2. Low-Power Hardware Platforms and Security

Several applications of security mechanisms targeting low-power embedded hardware platforms have been implemented. Junyoung et al. [20] performed security analysis on existing commercial platforms that are commonly used by many customers. These devices include Samsung smartphones, Synology Network Attached Storage (NAS) devices, smart televisions, several network routers, and some low-end sensing IoT devices measuring humidity and temperature. For most of the devices, the authors were able to extract the binaries from the platforms to reverse-engineer the code and extract encryption keys. Platforms implementing trusted execution environments known as the ARM TrustZone technology are more complex to reverse-engineer and are thus considered safer. In most of the cases, embedded security was not considered during the design phases of the platforms. Junyoung et al. [20] also proposed a secure design platform implementation, which should lead to more secure embedded IoT platforms. Their platform is based on the ARM Cortex-M23 architecture, which offers TrustZone, a secure bootloader mechanism, several cryptographic accelerators, and secure key storage. Their use case implements a smart plug monitoring the power consumption on a main socket and transmits the measured data via WiFi to a remote server. Before proper operation, the smart plug is first used for key creation followed by an authentication phase with a server. Then, the power of the socket is measured and sent to the server. Finally, a user can authenticate to the socket with the appropriate key. Although this scheme is simple and effective, no key refresh is performed during the life cycle of the smart plug.

Cryptographic algorithms play a very important role in transmitting data securely from one end-point to another end-point. However, many microcontrollers do not include cryptographic accelerators for data encryption and decryption. Mohammed El-hajj et al. [21] performed a comparison study of multiple lightweight NIST available software-based encryption and decryption schemes, both on Arduino Uno. For each of the methods, their results included the compiled code size in bytes required, the required power, the throughput, and the latency, from which a comparison against the AES-128-256 methods was drawn. The AES-128-256 scheme fits the available 32 kB of flash size of the Arduino Uno and can perform encryption and decryption at a rate of approximately 16 kBps. Other more lightweight schemes outperform the AES-128-256 scheme both in terms of data throughput and code size. However, these methods are much less commonly used in embedded IoT platforms. Although their study proves the feasibility of implementing lightweight software-based cryptographic solutions on constrained microcontrollers, care must be taken when a fully fledged IoT application is to be implemented on a microcontroller. Some applications might even demand much higher encryption and decryption throughputs, which require hardware-based cryptographic solutions.

Several IoT platforms are currently being proposed by different vendors. Many platforms are proposed with the aim of offering an open solution allowing for the development of any type of IoT application. Some platforms, however, target a specific range of solutions. In some cases, an operating system comes along with the hardware platform. The Zoul Zolertia ecosystem, proposed by Zolertia [22], is a small but versatile IoT platform offering multiple features. The platform is based on the Texas Instruments CC2538 system-on-chip (SoC), which is based on the ARM Cortex-M3 architecture. The platform offers wireless connectivity through the IEEE.802.15.4 standard which can be used to operate ZigBee, 6LoWPAN, and MiWi networks and runs the Contiki operating system [23]. The CC2538 offers hardware-accelerated AES128/256 encryption together with SHA128/256 hashing. While the platform has been very popular in research environments, it lacks support for secure boot and TrustZone. The Nordic Semiconductor manufacturer proposes several very popular SoCs geared toward IoT applications. An example is the nRF52840 [24], which offers the same IEEE.802.15.4 wireless connectivity as can be found for the Zolertia modules. In contrast to the Zolertia modules, the nRF52840 is based on the ARM Cortex-M4 architecture and features integrated cryptographic operations such as AES128 and SHA128/256. This platform also offers the TrustZone and secure boot and root of trust technology. A limitation, however, is the lack of support for secure peripherals such as anti-tamper input–output (IO) and the ability to select which peripherals belong to the secure and non-secure applications. nRF52840 SoC is also supported by the Contiki operating system [25]. Another very popular IoT development platform is based on the ESP32 modules manufactured by Espressif [26]. These integrated platforms typically offer wireless connectivity such as WiFi and Bluetooth Low Energy (BLE); several IO options; and many peripherals such as SPI, I2C, I2S, and UART. The most high-end version runs up to 240 MHz and can be programmed within the Arduino framework. Tailored toward performance sensing, these platforms lack the availability of cryptographic accelerators, TrustZone, and secure boot. Very popular among the maker community, Raspberry Pi is available in many flavors and with many options. Adafruit provides the Feather RP2040 board with an RF95 LoRa Radio transceiver [27]. This module offers several IO options together with some peripherals. RP2040 is a dual-core ARM Cortex-M0+ microcontroller. Although very popular, this microcontroller has no hardware support for security mechanisms. AREThOU5A [28] is an IoT platform specifically tailored toward water irrigation in agricultural applications. The platform is able to read out some analog and digital sensors for better control in irrigating plants. The platform is built around the ESP32 module. The Particle Photon 2 IoT Development Board shares the same form factor as the Feather board [29] and is based on the Realtek RTL8721DM microcontroller. This microcontroller is based on the ARM Cortex-M33 architecture running at 200 MHz and features TrustZone, secure boot options, on-the-fly encrypted firmware decryption, AES, and SHA support. Similarly to the Feather board, this platform is also tailored toward general IoT applications. Although this platform incorporates the TrustZone technology, no IO or peripherals can be directly dedicated to a secure or non-secure application. An alternative approach is offered by using direct memory access with interrupt options for a secure application.

### 2.3. Client–Server Security Protocols

For the security protocol, the description is restricted to symmetric-key-based protocols because we want to limit the impact of adding security to the system. One of the most prominent areas where symmetric-key-based protocols have been defined is in authentication and key agreement (AKA) protocols for fifth-generation (5G) networks [30]. Recently, Yadav et al [31] provided the latest result in this area. The scheme is shown to outperform other similar work [32,33,34,35] by providing more security features. More specifically, only [31,34] investigated protection against session temporary key attacks. In addition, only [31,35] investigated protection against perfect forward secrecy. However, this protection is not complete as their solution requires the storage of the two last keys in order to avoid desynchronization attacks. Therefore, the last session key is always vulnerable in case the private key material on the server side is leaked.

Also, outside the context of 5G-AKA, several symmetric-key-based schemes have been proposed in the literature. In [36], two highly efficient symmetric-key-based client–server schemes were proposed by relying on only one hash function: one with storage on the server side and one without storage. These schemes do not offer protection against perfect forward secrecy on the server side and are vulnerable to denial-of-service (DoS) attacks. The scheme is shown to outperform, both in performance and security, other symmetric-key-based client–server authentication schemes presented in [37,38,39,40]. All of them require a synchronized clock for proper functioning, and none of them offer protection against perfect forward secrecy. In addition, the systems in [37,39,40] are not resistant to ephemeral leakage attacks, where the temporary session key data are leaked.

Regarding PUF-based security protocols, a very interesting review is provided in [41]. Here, the author analyzed 15 PUF-based protocols published between 2016 and 2020. These protocols were divided into two categories according to their architecture: a client–server architecture, where only the client utilizes PUFs, and a device-to-device architecture where both possess PUFs. Regarding the first category, which is related to the focus of this paper, 11 papers were discussed. Four of them relied only on symmetric-key-based operations [42,43,44,45]. In [42], a PUF-based identity-preserving authentication protocol was proposed. This protocol is vulnerable to replay and spoofing attacks as no verification is performed on the server side to check if the random number sent by the device to the server is fresh. In addition, there is also no protection against anonymity and perfect forward secrecy. Moreover, scalability is an issue as it is not specified what needs to be done when the stored random numbers on the server are all used. In [43], an FPGA PUF-based authentication protocol for medical IoT devices was proposed to enable the one-way authentication of the medical device to the server. Also, this protocol is vulnerable to replay attacks and offers no protection against an insider attack. Since responses are sent in clear text, it will become possible to generate a PUF model of the device and thus impersonate the device. In [44], Nozaki et al. developed an XOR-arbiter PUF-based one-way authentication protocol that is resistant to machine learning attacks by relying on a secret sharing approach for hiding responses from outsiders. However, the protocol is not secure against insider attacks as the verifier can construct the challenge–response pairs and thus can impersonate the device. If the verifier is compromised, all pairs can be retrieved, and the protocol still becomes vulnerable to machine learning attacks. Moreover, the scheme is not scalable. In [45], a PUF-based authentication scheme for RFID systems was proposed. This scheme turns out to be vulnerable to synchronization attacks. Furthermore, the server is also able to construct a PUF model of the device as it has access to both challenge and response. As such, the scheme is also vulnerable to malicious insiders. Another PUF-based symmetric-key agreement scheme has been published in [46]. However, this scheme does not even satisfy the unlinkability feature and is also vulnerable to a malicious insider as the server is able to generate a PUF model. Note that only in the protocols of [42,43,45] is a real PUF implemented on PFGA. None of the schemes consider the inclusion of a PUF on a microcontroller-based device.

## 3. Preliminaries

We first explain the architecture of the system that we envision. Then, we describe the security risks in depth. Based on these risks, conclusions are drawn regarding device capabilities and the required security features in the authentication and key agreement protocol.

### 3.1. Architecture

The system we developed for sensing environmental variables consists of two major parts. On the one hand, a remote server ensures the proper collection of the sensed variables, while on the other hand, an embedded system (i.e., a client) captures the sensed values from several sensors. Urban environments typically provide many different methods for data communication such as the regular internet, wireless internet (WiFi), cellular communication networks, 4 G and more recently 5 G. While these communication channels typically tailor consumer electronics such as portable computers, smartphones, and servers, IoT devices, by contrast, rely on more power-efficient communication channels such as ZigBee, LoRa, 6LoWPAN, Bluetooth, Bluetooth Low Energy (BLE), etc. In our case, the communication between the remote server and the embedded systems is achieved by means of a Bluetooth Low Energy (BLE) connection. For environmental applications that do not require the transmission of large volumes of data, BLE can be utilized. However, more dedicated microcontrollers usually lack BLE functionalities. This shortcoming can be circumvented by adding a PCB add-on module such as the Microchip BM70/71 BLE [47], which communicates with the dedicated microcontroller via UART. The overhead of configuring this module can be minimized by initializing the hardware with a regular computer using the Microchip BLE configuration tool. The microcontroller is then able to transmit data over BLE via the standard UART interface with the desired specifications. BLE was chosen since many generalized IoT devices offer BLE connectivity (Section 2.2) together with other networking options such as ethernet, WiFi, etc.

Figure 1 highlights the concept of environmental sensing and transmission of the sensed variables to a remote server.

### 3.2. Security Risks

The concept of distributing many small embedded sensing IoT devices could facilitate decision-making regarding urban development. However, only by guaranteeing proper data collection and transmission can appropriate conclusions be drawn. Several security problems can occur during this process. In this section, we provide possible security issues that might occur during the lifetime of the deployed IoT devices. Alongside these issues, we also discuss the countermeasures that are implemented to overcome them.

Security threat: In case the device is physically accessed or manipulated by an unauthorized third party, the acquired sensor data can be read. The unauthorized third party can potentially access the key material and alter, remove, or add sensed data. It could use the stored key material to retrieve previous session keys and reveal data sent before the attack.Countermeasure: It is important to encrypt data using a key, which is generated by means of stored data and a PUF. If a device is tampered with, the output of the PUF is changed, and no valid key will be generated anymore. Moreover, the key management protocol should satisfy perfect forward secrecy such that if key material is revealed somehow, it is impossible to retrieve previous session keys and generate appropriate hashes of all stored data on the external flash memory.Security threat: An attacker attempts to clone the device, for which the corresponding measurements can then be manipulated.Countermeasure: Integrating the PUF in the authentication and key establishment mechanism will avoid device spoofing as the PUF is unique for each device.Security threat: The device can be moved to an undesired location, resulting in wrong measurements of the environmental variables.Countermeasure: The counter-proposition is to include readings from an integrated GPS module to avoid collecting measurements from unexpected locations. The transmitted data should then be encrypted, resulting in the protection of their integrity, so it is impossible to change the GPS values. The GPS is connected to a secure application residing in the TrustZone area. The location can still be accessed by a non-secure application via a special function that calls to the secure application.Security threat: The wireless communication channel allows third parties to access, alter, jam, block, or spoof (parts of) the transmitted data, corresponding to an active and passive attacker profile in the literature [48].Countermeasure: The communication between the edge device and the remote data collection point should be encrypted. Jammed transmitted data are stored locally until successful transmission. Keys are regularly updated. It is also advisable that the communication is anonymous and non-linkable, such that no particular sensor is envisaged for dedicated attacks. Special attention should be given to desynchronization attacks, which have the aim of destabilizing the relationship between client and server such that no future keys can be agreed upon.Security threat: Trusted third parties (TTPs), who are responsible for distributing the key material, might be interested in obtaining the data themselves for their own purposes, e.g., selling the data.Countermeasure: It is important to have a strict separation between the registration phase (in which TTPs are involved) and the initialization phase (which is only between the device and the server).

### 3.3. Security Protection Mechanisms

Based on the risks discussed above, the following conclusions are drawn regarding device capabilities and the security features of the authentication and key agreement protocol:

#### 3.3.1. Device Capabilities

Small, low-cost, low-power, and resource-constraint embedded sensing devices are typically used to monitor many environmental variables. The capabilities of these devices are generally too restrictive to implement standard security features and protocols to counter possible threats found in modern computer technology. Moreover, standard protocols like TLS do not cover all the required security features mentioned above (e.g., anonymity). As such, a dedicated low-cost solution needs to be implemented that exploits the available resources on these devices, including inherent hardware security mechanisms, as much as possible. For instance, at the very least, tamper-resistant protection and the presence of a PUF mechanism are required. In addition, hardware-enabled accelerators such as encryption and decryption, hashing methods, transmission options, secure key storage, etc., are selected to assist the operation of embedded platforms.

#### 3.3.2. Security Protocol

Standard security in BLE does not offer all the security features mentioned above. Therefore, a dedicated security protocol is required. From the security risks mentioned above, we can conclude that, besides classical features such as confidentiality and mutual authentication, the authentication and key agreement protocol should also satisfy anonymity and unlinkability, perfect forward secrecy, and protection against desynchronization and denial-of-service attacks.

## 4. IoT Hardware for Environmental Monitoring

### 4.1. Hardware Architecture

Our hardware platform is developed around an ARM-Cortex M23-based low-power PIC32CM5164LS00064 microcontroller [49] manufactured by Microchip. This microontroller offers the required security mechanisms, as further discussed in Section 4.2.

The hardware also provides sensors to measure temperature, relative humidity, perceived light intensity, perceived acoustic intensity, and acceleration. It is also possible to retrieve the current location of the platform in outdoor settings via the implemented L96-M33 GPS receiver.

The hardware is able to communicate to a remote computer or server via BM70 [47] BLE connectivity. The microcontroller also enables direct USB communication with a computer via a UART-to-USB converter for debugging and initialization. However, the debugging and initialization port is not intended for use during normal operation. Figure 2 shows an overview of the implemented hardware.

### 4.2. PIC32CM5164LS00064 with ARM-Cortex M23 and TrustZone

The PIC32CM5164LS00064 microcontroller is based on the ARM-Cortex M23 and possesses many hardware-enabled security features such as (but not limited to) the following:Data privacy via encryption methods, including AES-128/192 and 256 (CBC mode), GCM, and data authenticity via hashing methods such as SHA-128, 256, and 512.A true random generator that is based on a silicon or device-specific entropy source and is guaranteed to be unique and unclonable for each device.Trusted peripherals: many peripherals, including the many communication buses required by sensors, the real-time clock (RTC), the different clock mechanisms, and some input–outputs (IOs).SHA- or HMAC-based secure boot that enables the authentication of the user boot loader image. When the authentication fails, the microcontroller reinitiates the boot process until successful authentication.Trusted execution environment (TEE) via TrustZone: TrustZone is a hardware-assisted TEE and allows one to compartment the application in secure and non-secure code sections, mitigating the risks of malicious firmware execution. The microcontroller first initiates the secure application before the non-secure application counterpart is booted. The programming tools provided by Microchip allow us to customize the allocated resources for each of the applications. This allocation includes the following features:-The amount of flash memory and volatile memory (RAM);-The selection of peripherals (I2C, SPI, and UART) and input–output (IO) ports;-Internal modules such as timers, RTC, etc.Data flash with tamper-proof option: up to 16 kB of the flash memory can be allocated to the trusted execution environment to store configuration and sensitive application data and can optionally be configured to be tamper-proof.

Figure 3 depicts the organization of the application on the PIC32CM5164LS00064 microcontroller. The application consists of three major parts residing in a dedicated memory segment. On the one hand, the secure application resides in TrustZone. This application is written in C and manages all secure peripherals, secure input–output (IO), secure boot options, and optionally the data flash that is allocated to TrustZone. On the other hand, the non-secure application part is designed to house most of the application. From a firmware point of view, this application resembles the firmware of microcontrollers, which do not offer the TrustZone feature with the exception of stringent security features that are housed in the TrustZone area. This application can be written in both C and C++ and also houses libraries such as the WolfSSL cryptographic library. Non-secure peripherals are also allocated to this region. A third small portion of the resources is allocated to the veneer area. This area acts as a demilitarized zone and allows the non-secure application to gain indirect access to secure features via dedicated function calls. During each call, the veneer function must assert that the provided function arguments are valid and that the provided pointers indicating data buffers reside in the non-secure application area. Each of the regions can be allocated a given amount of code ROM, data flash, and volatile memory (i.e., RAM). This allocation can be adapted to the needs of the application and is set via dedicated registers in the microcontroller. Figure 3 depicts typical amounts of each type of memory that can be allocated to each of the areas. The veneer function area is usually allocated the least number of resources since veneer functions bridge the communication between the non-secure application and the TrustZone area.

### 4.3. Physical Unclonable Function

The low-power ARM-Cortex-M23 and ARM-Cortex-M33 offer very similar security features. The architecture is implemented by many manufacturers into a silicon-integrated circuit to which several features are added. One of these features is the SRAM-based physical unclonable function (i.e., PUF), which can be made accessible in a similar fashion as most other integrated peripherals such as SPI, I2C, etc. The SRAM-based PUF guarantees the ability to generate a certain value (i.e., response) based on a given input value (challenge). The small differences in silicon implementation guarantee that each manufactured microcontroller produces different but stable outputs. NXP offers the LPC5500 microcontroller series [50] implementing the SRAM-based PUF technology. Although this technology would be very beneficial in the case of an SRAM-PUF-based challenge–response algorithm for new secret key generation, the lack of availability of this feature in all other currently available microcontrollers based on the same architecture would restrict the proposed protocol only to the LPC5500 series of microcontrollers. Most of the ARM-Cortex-M23 and ARM-Cortex-M33 systems, however, do offer a true random generator based on an entropy source, which is unique to each manufactured device. An alternative PUF method can be implemented by utilizing this true random number generator and calculating a PUF-similar function in the TrustZone area. The repeatability of the PUF can be guaranteed by storing the generated random number in the secure flash of the microcontroller, which can be made accessible to the TrustZone code only.

## 5. Key Agreement Protocol for Generating New Symmetric Secret Keys

Here, we first describe the notations used in the security protocol, followed by a detailed description of each of the steps included in it.

### 5.1. Notations

The protocol utilizes several cryptographic operations that are available in the microcontroller. In the remainder of this paper, the following cryptographic operations are presented:A hashing function H() accepts a variable input length and provides a digest that can be used for authentication. The PIC32CM5164LS00064 microcontroller allows us to compute hashes of 128, 256 and 512 bits in length;An exclusive or (XOR) operation is used and denoted via the ⊕ operation;Several operations involve the concatenation of multiple values, denoted as |;An SRAM-based PUF mechanism is presented, expressed as PUF(). As described above, many microcontrollers do not offer an SRAM-based PUF mechanism. Therefore, we mimic this behavior by implementing an alternate method in the TrustZone area based on a preloaded true random number $U_{init}$ generated by the true random number generator stored in the secure data flash. Via the hashing functions implemented in the microcontroller, it is possible to compute the hash of a given key $K_{i}$ and a challenge $C_{i}$, together with this preloaded true random value. In the remainder of the paper, we the response $R_{i}$ is computed using $H\left(K_{i}|\left(U_{init}|C_{i}\right)\right)=R_{i}$.

### 5.2. Authentication and Key Agreement Protocol

The firmware is implemented on the microcontroller, while a computer is used as the server. The key agreement protocol is implemented via three different state machines:Registration;Initialization;Authentication and key agreement.

Registration (1) and initialization (2) are only issued once, while the authentication and key agreement (3) phase is issued at a regular time interval. Phase 1 requires the interaction of a trusted third party (TTP). This TTP can either communicate with the embedded microcontroller (the client) via a direct physical connection or via a Transport Layer Security (TLS) connection. Aside from the different states, the computer (i.e., server) can always request from the microcontroller (i.e., client) to report in which state it currently is. This is especially important to detect if the registration and initialization phases have already been executed. In case these phases are not successfully completed, the two parties are not allowed to proceed to the third phase.

#### 5.2.1. Registration

For the registration phase, the TTP is involved, and a secure channel is assumed between the TTP and the server (e.g., pre-established TLS connection channel) on the one hand and the TTP and the client (e.g., physical presence) on the other hand. The TTP first validates the identity $ID_{i}$ of the client, and if successful, a common key $K_{i}$ is shared with the server and the client. On both the client and server side, the pair $\left(ID_{i},K_{i}\right)$ is stored.

We now describe the registration phase in more detail. This phase consists of exchanging the $ID_{i}$ and a common key $K_{i}$ between both the computer of the TTP and the microcontroller. The state machine of the registration phase performs three major tasks:Collecting the $ID_{i}$ of the microcontroller: each microcontroller contains a 128-bit unique serial number, which can be used as $ID_{i}$ and is stored in the read-only address 0x0080A00C of the microcontroller. Once a physical connection between two entities is established, the computer requests the $ID_{i}$ from the microcontroller. This $ID_{i}$ can be stored in a secure database for later communication.Key $K_{i}$ generation: after the successful transmission of the $ID_{i}$, a 128-bit long key $K_{i}$ is generated on the computer at the TTP and transmitted to the microcontroller. This key is generated by hashing (i.e., SHA-512) the current timestamp $T_{c}$, which is concatenated with a random value $U_{c}$ of 128 bits so that $K_{i}=H\left(T_{c}|U_{c}\right)$. The microcontroller acknowledges the proper reception of the key. The microcontroller stores the key $K_{i}$ in the secure data flash.The final step of the registration phase consists of generating the $U_{init}$ true random value and also storing it in the secure data flash. Both the random value $U_{init}$ and the key $K_{i}$ can only be accessed via the TrustZone area.

The registration state machine is shown in Figure 4.

#### 5.2.2. Initialization

The initialization phase follows a similar flow as described for the registration phase. During this state, the same physical connection between the server and the client is established through a secure connection. At first, the server generates a random 256-bit long challenge $C_{i}$, which is transmitted to the client. The client accepts the challenge and computes the response $R_{i}$ via $H\left(K_{i}|P\left(C_{i}\right)\right)=R_{i}$. This phase allows us to confirm the proper functioning of the entire challenge–response chain. In case the computation of the response fails, the server is notified, and the client can then be refused. Otherwise, the client acknowledges the proper generation of the response $R_{i}$ by transmitting the response to the server. The server then stores the resulting challenge–response pair $\left(C_{i},R_{i}\right)$ in a database. The same challenge–response pair $\left(C_{i},R_{i}\right)$ is stored in the secure data flash memory of the microcontroller. The initialization phase’s state machine is shown in Figure 5.

#### 5.2.3. Authentication and Key Agreement

Once both the registration and the initialization phases have been successfully completed, the embedded device enters the authentication and key agreement phase. The server initiates the authentication and key agreement phase at regular time intervals, which can be adapted following specific requirements. To start this process, the server transmits a specific message to initiate the process. The server polls the client with this message as long as no reaction is received from the client. As the client is expected to be deployed on a remote location, connection disturbances issues might occur, causing the two end-parties to be temporarily disconnected. The client might also not be receiving or transmitting as it is temporarily in a sleep state to reduce energy consumption. At the reception of the start message, the client starts the process by generating a random 128-bit variable $U_{1}$ after which a 512-bit hash function $H(U_{1}|C_{i}|R_{i}|K_{i})\;=\;(v_{1}|v_{2}\left|v_{3}\right)$ is computed. The output $\left(v_{1}|v_{2}|v_{3}\right)$ yields the concatenation of $v_{1}$, $v_{2}$ and $v_{3}$, with a bit-width of 128, 256, and 128 bits, respectively. The output $v_{1}$ is used to derive a new challenge $C_{i+1}$ and new key $K_{i+1}$ with $H\left(U_{1}|v_{1}\right)=\left(C_{i+1}|K_{i+1}\right)$. The newly generated key and challenge are both 256-bit wide, resulting in a 512-bit wide hash. In our case, the new key $K_{i+1}$ is truncated to 128 bits to encrypt messages between the client and the server. With the newly obtained challenge $C_{i+1}$ and new key $K_{i+1}$, we derive the new 256-bit-wide response $R_{i+1}$ via $H\left(K_{i+1}|P\left(C_{i+1}\right)\right)=R_{i+1}$. The microcontroller generates a first message $M_{1}$ to be transmitted to the server, which consists of the random value $U_{1}$; the new challenge $C_{i+1}$; the last part $v_{3}$ of $\left(v_{1}|v_{2}|v_{3}\right)$; and the hidden response $d_{i}$, which is computed via $R_{i+1}⊕v_{2}=d_{i}$ so that $M_{1}=\left\{U_{1},C_{i+1},v_{3},d_{i}\right\}$.

Upon the arrival of the message $M_{1}$, the random value $U_{1}$ is extracted from the message, and the server also computes $H(U_{1}|C_{i}|R_{i}|K_{i})\;=\;(v_{1}^{*}|v_{2}^{*}\left|v_{3}^{*}\right)$, using the current key $K_{i}$ and response $R_{i}$, which belong to the current challenge $C_{i}$. In case the comparison $v_{3}^{*}==v_{3}$ is true, the server accepts the message $M_{1}$ and computes the new challenge $C_{i+1}$ and response $R_{i+1}$ via $H\left(U_{1}|v_{1}^{*}\right)=\left(C_{i+1}|K_{i+1}\right)$. The corresponding new response $R_{i+1}$ can be retrieved by reapplying the XOR operation on $d_{i}$ so that $d_{i}⊕v_{2}^{*}=R_{i+1}$. At this stage, the old challenge $C_{i}$, the key $K_{i}$, and the response $R_{i}$ are replaced with their new counterparts $C_{i+1}$, $K_{i+1}$, and $R_{i+1}$.

A confirmation message $M_{2}$ for the successful reception of $M_{1}$ is transmitted to the client by generating a dedicated session key SK. To achieve this step, the server first generates a new 128-bit random generator $U_{2}$. A session key $SK=v_{4}$ is computed by applying a 512-bit hash so that $H(U_{1}⊕U_{2}|C_{i+1}|R_{i+1}\left|,K_{i+1}\right)=\left(v_{4}|v_{5}\right)$. In this process, the XOR operation between the random number $U_{1}$ from the client and $U_{2}$ from the server is taken. This value is concatenated with the newly generated values of $C_{i+1}$, $R_{i+1}$ and $K_{i+1}$. The confirmation message $M_{2}$ consists of the newly generated random number $U_{2}$, followed by $v_{5}$, so that $M_{2}=\left\{U_{2},v_{5}\right\}$.

Upon the arrival of this message, the value $v_{5}$ is extracted by the client, and the same hash $H(U_{1}⊕U_{2}|C_{i+1}|R_{i+1}\left|,K_{i+1}\right)=\left(v_{4}^{*}|v_{5}^{*}\right)$ is computed. In case the comparison $v_{5}^{*}==v_{5}$ is true, the session key on the client side equals $v_{4}^{*}$, while the old challenge $C_{i}$ and the key $K_{i}$ are, respectively, replaced with their new counterparts $C_{i+1}$ and $K_{i+1}$. The whole mathematical operation flow is described in Figure 6, while Figure 7 depicts the corresponding state machines.

## 6. Security Analysis

### 6.1. Informal Security Analysis

We first explain how the system informally satisfies the required security features mentioned in Section 5.

Confidentiality: Since an attacker has no access to $R_{i},K_{i}$ or $R_{i+1},K_{i+1}$, it is impossible to derive the common shared key that is required as input for the hash function.Mutual authentication: The server is able to uniquely authenticate the client as only the client is aware of $R_{i},K_{i}$ and is able to derive a valid value for $v_{3}$. The same holds for verification by the client by means of $v_{5}$. Freshness is achieved thanks to the usage of random values $U_{1}$ of the client and $U_{2}$ of the server.Anonymity and unlinkability: The only information that uniquely identifies the client is $C_{i}$, which is not directly linked to the identity, and thus anonymity is obtained. Due to the fact that $C_{i}$ updates after each successful authentication session, session unlinkability is achieved.Protection against ephemeral secret leakage: In this case, there is no secret ephemeral key material as the randomly generated values $U_{1},U_{2}$ are also sent in the public channel.Perfect forward secrecy: All keys used in the scheme are temporal and are updated after each successful session by means of a one-way hash function. Even if $K_{i+1}$ is leaked, it is impossible to derive the previously generated session keys as they rely on knowledge of the previous common key $K_{i-1}$, which can not be gained without breaking the hash function.Protection against desynchronization attacks: Suppose that the attacker intercepts $M_{1}$; then, the attacker will never be able to send a valid $M_{2}$ message, and the client will not update its key material. If $M_{2}$ is intercepted, the server updates its key material, but the client still stores both the old and the new values for $K_{i},C_{i}$. It is only if it has received a valid value for $v_{5}$ that the change is made.Protection against DoS attacks: Since there are only two communication phases, and each entity can directly check the validity of the message, there are no requests that need to remain open for longer periods, limiting the success probability of a DoS attack.Secure usage of PUF: The scheme does not suffer from PUF spoofing since the generated PUF responses are uniquely linked to the server, including the common shared key. Therefore, the server is not able to abuse the PUF responses to impersonate the device. As the communicated responses do not include the direct PUF outcome and are constructed through a hash function, it is not possible to use them in a machine learning model.Semi-trusted TTP: If the registration and initialization phases are organized independently of each other, and the initialization phase is established, for instance, through physical presence, the TTP is not aware of the commonly agreed challenge–response data. As such, the TTP will not be able to follow the communication between the two further and will only play a role during the registration phase to validate the legitimacy of the communication entities.

### 6.2. Formal Security Analysis

In this section, we provide the formal proof using the Real-Or-Random (ROR) model, which is common in the literature. This model is developed according to the capabilities of the attacker, described in Section 3.

The participants P in our scheme are the client Cl; server Se; and random oracle *O*, denoted by P={Cl,Se,O}. We need to prove that only Cl,Se are able to derive the common session key SK. If so, Cl,Se are called partners and the SK fresh.

In the attack model, the following queries can be made by the attacker A:Hash queries H(m): for each newly submitted value of *m*, an entry (m,H(m)) will be added to the list $L_{H}$;PUF queries P(C): for each newly submitted challenge *C*, an entry (C,P(C)) will be added to the list $L_{P}$;Execute queries, Execute($M_{1})$ and Execute($M_{2})$: these queries reveal the content of the sent messages M1 and M2, respectively, and thus correspond to a passive attack;Send queries, Send$\left(Cl,M_{1}\right)$ and Send$\left(Se,M_{2}\right)$: these queries allow an attacker to intercept, change, insert, and delete (parts of) the content of the messages $M_{1},M_{2}$ and thus correspond to an active attack;Corrupt queries Corrupt(A): these queries enable an attacker to derive the long-term secret key material $K_{i}$, stored both at the server and the client. Extracting the PUF output without destroying the device and, thus, the long-term key $K_{i}$ is impossible;Test queries T(A): if the output of the flipped coin equals 0, a random value is returned, and otherwise the SK. The goal of this query is to verify if the attacker is able to distinguish a random value of the correctly guessed SK.

Denote the probability that the attacker wins the game as Pr[succ]. Then, the advantage of the attacker A is defined by Adv(A)=|2Pr[succ]−1|. If Adv(A)<ϵ, for any sufficiently small ϵ≥0, the semantic security of the proposed scheme is broken.

**Theorem** **1.**
*Let A be a polynomial time attacker against the semantic security of our proposed scheme; then, the advantage of A is bounded by*

(1)
$$\begin{array}{c} Adv\left(A\right)\leq \frac{q_{h}^{2}}{2|Hash|}+\frac{{\left(q_{s}+q_{e}\right)}^{2}}{2|Hash|}+\frac{q_{p}^{2}}{\left|PUF\right|} \end{array}$$

*where $q_{s},q_{e},q_{h},q_{p}$ represent the number of send, execute, hash, and PUF queries, and |Hash|,|PUF| denote the range space of the hash and PUF function, respectively.*


**Proof.** As proven in [51,52], the success probability of the attacker only increases by a negligible amount when moving from one game to the other. This principle is called game hopping and is used in this proof. We propose four games $\left\{G_{1},G_{2},G_{3},G_{4}\right\}$ and denote the corresponding success of the attacker A in each of these games by $succ_{i}\left(A\right)$ for i∈{1,…,4}.
Game $G_{1}$: The first game represents the direct attack, without the use of any of the queries defined above. By definition, we have
(2)$$\begin{array}{c} Adv\left(A\right)=2Pr[succ_{1}\left(A\right)-1] \end{array}$$Game $G_{2}$: In the second game, the attacker eavesdrops on the communication channel through the execute queries to retrieve $M_{1},M_{2}$. However, these messages do not include details on the SK, and thus, the attacker does not have an additional advantage in order to reply to the test query. As such,
(3)$$\begin{array}{c} Pr\left[succ_{2}\left(A\right)\right]=Pr\left[succ_{1}\left(A\right)\right] \end{array}$$Game $G_{3}$: In the third game, the attacker applies the send, hash, and PUF queries before consulting the test query. Due to the birthday paradox, the probability of collisions between the outputs of a hash function is less than $\frac{q_{h}^{2}}{2|Hash|}$; between the outputs of the PUF, it is less than $\frac{q_{p}^{2}}{2|PUF|}$; and between the content of the messages, it is less than $\frac{{\left(q_{s}+q_{e}\right)}^{2}}{2|Hash|}$. Therefore, we have
(4)$$\begin{array}{c} |Pr\left[succ_{3}\left(A\right)\right]-Pr\left[succ_{2}\left(A\right)\right]|\;\leq \;\\ \frac{q_{h}^{2}}{2|Hash|}+\frac{{\left(q_{s}+q_{e}\right)}^{2}}{2|Hash|}+\frac{q_{p}^{2}}{2|PUF|} \end{array}$$Game $G_{4}$: In the last game, the attacker executes all possible queries, including corrupt queries to the client or the server. However, as the knowledge of the outcome of the PUF is also required, the probability of success is restricted to the probability of a collision in the PUF outcome. As such, the following holds:
(5)$$\begin{array}{c} |Pr\left[succ_{4}\left(A\right)\right]-Pr\left[succ_{3}\left(A\right)\right]|\leq \frac{q_{p}^{2}}{2|PUF|} \end{array}$$
Due to the difference lemma [51], we can combine Equations (Equation 2)–(Equation 5), resulting in Equation (Equation 1). □

### 6.3. Comparison of Security Features

Table 1 compares the security features for the different relevant and related symmetric-key-based client–server schemes available in the literature.

As can be seen, none of the PUF-based key agreement protocols satisfies the anonymity and unlinkability security feature. Only a very limited number of schemes are resistant to insider attacks, which means that the server is not able to take over the role of the device. This follows from the fact that the server shares the same symmetric key in [31,36] or receives enough data on the challenge–response pairs in clear text such that it is able to generate a machine learning model to simulate the PUF as in [43,44,45,46]. Often, the PUF-based schemes in the literature are not scalable when parts of challenges and responses are sent in clear text [42,43,44,45]. Moreover, all of the PUF schemes implemented in real time [42,43,45] in the protocols are on FPGA, and none of them involves a microcontroller as in this scheme.

## 7. Performance Analysis

The different security solutions proposed in this paper have a certain impact on the overall performance of the embedded system and the intended application. In this section, we provide an overview of the impact of these solutions in terms of the additional memory, peripheral, and CPU time requirements to perform these tasks. Firstly, in Section 7.1, the impact of using TrustZone on the available code flash, data flash, and volatile memory is depicted. In the second phase, we evaluate the impact of using TrustZone in terms of required additional CPU time. This analysis includes the time required to transfer a given amount of data from the non-secure application to the TrustZone area via the veneer functions. Secondly, in Section 7.2, the impact of the different encryption and hashing methods on the communication with the remote server is analyzed. We evaluate the AES CBC-128 and AES GCM-128 encryption methods and the SHA-256 hashing method in terms of required CPU time and memory footprint. To assess the overall performance regarding most of the typical packet sizes that are used in our environmental monitoring applications, the required CPU time performance is evaluated on different input data size lengths. Lastly, in Section 7.3, we evaluate the requirements for the key agreement protocol in terms of both the required memory and CPU time. All experiments were performed on the same platform with a clock speed of 48 MHz and running at a high-performance mode (i.e., “PL2”). All the implemented code was compiled using the “O2” compiler flag for C en C++. This flag does not provide the highest optimization levels but is the highest available optimization level provided by the free XC32 compiler.

### 7.1. Performance Impact due to TrustZone

The concept of TustZone allows one to dedicate a portion of the application to a secure area. This area is solely involved in the processing of security elements such as secure boot and the processing of secure peripherals. The regular application (non-secure application) is in charge of processing the regular application flow. The secure and non-secure applications flows, however, run in their respective memory spaces. Via special “veneer” functions, it is possible to call secure functions from non-secure applications. These functions can perform requests on secured elements. Via dedicated integrated checks, the validity of these checks can be guaranteed. It is, however, still the responsibility of the secure application developer to ensure the proper handling of the requests. The subdivisions of the applications are shown in Figure 8.

#### 7.1.1. Memory Allocation

Microchip allows us to allocate various parameters such as code flash size, volatile memory (RAM), and data flash for each of the applications. PIC32CM5164LS00064 offers 512 kB of code flash, 64 kB of RAM, and 16 kB of data flash that can be allocated to either application. In our case, we opted to allocate 256 kB (i.e., 50%) of code flash and 32 kB (i.e., 50%) of RAM to the non-secure application. The secure application utilizes up to 254 kB (i.e., 49.6%) of code flash and 32 kB (i.e., 50%) of RAM. The secure application is configured to handle 100% of the data flash (i.e., 16 kB). A small portion of the code flash, about 2 kB (i.e., about 0.4%), is allocated to the secure callable functions (i.e., veneer functions). The amount of memory utilized by the non-secure application inside the allocated memory space accounts for 207 kB (i.e., 80%) of code flash and 16.64 kB (i.e., 52%) of RAM. The secure application utilizes much fewer resources, with only 15 kB of code flash memory (6%) and 500 bytes (i.e., 3.1%) of RAM. In this case, the secure application is only in charge of performing the boot operation of the non-secure applications and processing the real-time clock and GPS.

Although most of the memory remains unused, care must be taken in order to prevent both applications from stalling due to out-of-stack space or out-of-code-flash space. Even though about 50% remains unused, the overhead of TrustZone has a severe impact on memory allocations, which results in the inability to utilize the entire code flash, RAM, and data flash for the application.

#### 7.1.2. Veneer Functions Calling Overhead

The non-secure and secure applications exchange multiple types of information, including the location (via the GPS) of the device, the current time (via the GPS and real-time clock (RTC)), and the data for the key agreement protocol. In our application, we opted to gather the data from the secure application via byte array structures, passed as pointer arguments to veneer functions. Aside from the pointer argument, the length of the byte array is provided as a second argument.

In all cases, the validity of the arguments and the pointers to the arrays of bytes must be checked. The pointer passed in the veneer function must point to a memory portion located in the non-secure area, while all the bytes inside this array must fit the same non-secure area. In our tests, we measured the overhead in terms of required clock ticks to the required amount of data from the non-secure to the secure application and vice versa. We performed three tests to estimate the impact of each of the operations, which were as follows:Test 1: measuring the impact of only calling the veneer function without any further processing;Test 2: measuring the impact of calling the veneer function with the pointer validity check;Test 3: measuring the impact of calling the veneer function with the pointer validity check and data copying to the secure application (or from the secure application into the array).

In the case of the first two tests, the overall overhead time of calling the veneer function and performing the pointer validity check remained constant with 104 CPUticks and 270 CPUticks, respectively. In test 3, the data were also copied between the two applications, which involved a linear time operation. We performed tests in which we transferred data in multiples of 16 bytes from the non-secure application to the secure application and vice versa. We started with 16 bytes and continued until reaching a data packet of 80 bytes. Each experiment was repeated 40 times, and the results are shown in Table 2.

All results provide the same timing for each transmitted length of bytes. Therefore, no standard deviation could be computed. The results show that for each additional 16 bytes to be transferred, another 625 CPUticks are required. The results also show that for low amounts of data to be transferred, i.e., only a few occurrences per second are required, meaning that the time impact of the transfer remains very limited to about 78μs for 80 bytes.

### 7.2. Overhead Induced by the Encryption and Hashing of the Messages

Encryption and hashing are useful to, respectively, hide the data and check the integrity of the transmitted data. Both methods, however, induce a given amount of overhead on the embedded platform in terms of the required CPU processing time, volatile memory, and code size (flash size). Both encryption and hashing are performed in the non-secure application. In our case, we used the AES-128 CBC method in conjunction with the SHA-256 hashing method and the AES-128 GCM encryption method. All cryptographic operations are readily available via the WolfSSL library provided by the manufacturer for this microcontroller.

#### 7.2.1. Transmission Length

The acquired sensory information is first serialized into a serializer before it is transmitted to the server. The serializer makes sure the packet containing the information also contains the required information for the other party to have the ability to process the data. Therefore, a header of 20 bytes is provided at the front of the package. The payload usually contains between 14 and 60 bytes. The length of the payload varies based on which sensors are read out, which operations have been performed, the current status, etc. The total packet length thus varies between 38 and 84 bytes. To apply encryption and hashing, the message is extended with the following bytes:AES-128 CBC together with SHA-256: AES-128 CBC requires the data to be encrypted and provided in multiple of 16 bytes in length. Therefore, zero padding with a length between 0 and 15 bytes is applied to the message. The AES-128 CBC method also requires an initialization vector (IV) of 16 bytes to be prepared and passed along with the message. Both the zero padding and the IV together lead to an additional message length increase between 16 and 31 bytes. The SHA-256 method computes a digest of 32 bytes of the complete message, which is added to the message, so the receiver can compute the integrity of the message. The total length of the original message is thus extended with a minimum of 48 and a maximum of 63 bytes.AES-128 GCM: AES-128 GCM performs both encryption and hashing during the cryptographic operation. Therefore, no additional hashing on the message is required. The AES-128 GCM method requires an IV of 96 bits (i.e., 12 bytes) and an additional 32 bytes for authenticity. Contrary to the AES-128 CBC method, no zero padding is required, resulting in an added length of 44 bytes to the original message.

In the case of short messages, the additional message overhead induced by the AES-128 CBC and SHA-256 methods leads to an overhead of up to 50 additional bytes for 14 data bytes. For long messages, however, both the AES-128 CBC and SHA-256 methods create an overhead of 52 bytes for a message containing 60 bytes. The latter method leads to a total packet length of 136 bytes to be transmitted compared to the original 84 bytes. With the AES-128 GCM method, however, the transmission would require an additional length of 44 bytes in both cases, leading to a total length of 84 and 128 bytes, respectively.

#### 7.2.2. Flash and RAM Usage

The AES and SHA functionalities are provided by Microchip by means of the WolfSSL library [53]. This library consists of many different encryption and hashing functions, and depending on the current microcontroller, several functions can be enabled. The Microchip MPLab X IDE programming tool also allowed us to select hardware support for the methods we chose. We opted to enable support for the standard AES methods together with the SHA family. The footprint in terms of volatile memory usage (RAM) and code usage is given in Table 3. As the library can only be implemented in the non-secure application, the results are related to the non-secure available RAM and flash.

The non-secure application is allocated to 50% of the resources of the microcontroller for both RAM and code flash. The results show that the cryptographic library takes up to 7.7 kB of code flash and 1.86 kB of RAM, leaving ample room for other codes and libraries.

#### 7.2.3. Required CPU Time for Encryption and Hashing

Aside from the transmission length overhead and the usage of code flash and RAM, the process of encryption and hashing takes a given amount of time to complete. The microcontroller provides three different hardware timers, which can be used to measure the time between two intervals. During our experiments, we selected “Timer0” to measure the time a given function needs to process. This timer provides a single counter register, which can be reset at any moment and increments at each CPU tick. Once the measurement of the function is completed, that register is read out, and the value is transmitted to the computer via the function, debugging port. During the experiments, an array with increasing lengths between 23 and 152 bytes containing dummy data was generated. Each of the operations was repeated at least 100 times, from which the average and the 95% confidence intervals were taken. The AES-128 encryption methods were applied to the useful payload only, leaving the 20 bytes of the header unencrypted. The SHA-256 hashing method, by contrast, was used on the whole packet, including the header of 20 bytes. Figure 9 shows the number of ticks for both the AES-128 and SHA-256 methods. In the case of the AES-128 encryption method, the time taken to encrypt a given message increased per multiple of 16 bytes in length. The SHA-256 hashing method showed a similar staircase scenario where the time required to compute the digest increased per multiple of 64 bytes. The required computing time of the SHA-256 hashing method remained at approximately 288 μs (i.e., 13,800 CPUticks) until 35 useful bytes were hashed (i.e., 55 bytes including the header). From 36 bytes, the required time to hash the payload increased by approximately 288 μs per each additional block of 64 bytes.

In all cases, the microcontroller operated at a clock frequency of 48 MHz. In the case in which a useful payload of 100 bytes was to be transmitted to the server, the microcontroller would require approximately 1.98 ms of computing time for the AES-128 CBC method together with SHA-256, while 3.16 ms would be required to compute with the AES-128 GCM method, resulting in a maximum transfer speed of 50 kBps and 31 kBps, respectively.

In all cases, the 95% confidence interval remained below 1500 CPUticks, providing a time deviation of up to 31.25 μs.

### 7.3. Key Agreement Protocol Overhead

Here, we compare the communication and computation overhead of our authentication and key agreement protocol with other related and recent symmetric-key-based protocols. Note that we did not include the schemes [42,43,45] as they are not scalable and not sufficiently secure.

For having a fair comparison among the different schemes in defining the number of bits sent, we considered the length of identity-related information to be 128 bits, with the outcome of hash operations set at 256 bits, timestamps at 32 bits, symmetric keys, and PUF challenges–responses at 128 bits. As can be seen from Table 4, our scheme can also be limited to a minimum of two phases, which is better than the other PUF-related schemes. With respect to the number of bits sent, our scheme does not outperform but is also not among the worst and is only slightly higher than the other existing schemes [36,42].

The key agreement protocol also generated a certain time overhead when computing the different steps. The overhead of performing the key agreement protocol was estimated by running the method 60 times, from which the average and 95% confidence intervals were computed considering the most impacting operations (Section 5.2.3). The results are listed in Table 5. Storing the newly generated secret key $K_{i+1}$, the challenge $C_{i+1}$, and the response $R_{i+1}$ together required approximately 2.3 ms and accounted for about 55% of the total amount of required time to perform the key agreement protocol. The generation of the random value $U_{1}$ always took a fixed amount of 11 μs, while computing a new response $R_{i+1}$ required approximately 635 μs. Next to storing the newly generated key, challenge, and response, the computation of the session key $SK=v_{4}^{*}$ with the related hash function required the most amount of CPU time, with approximately 850 μs. The total requested time to perform a full new key agreement took less than 5 ms. All the operations were divided into different stages, which individually required less than 1.25 ms to be performed.

To compare the computational overhead, we focus on the number of the most compute-intensive operations while disregarding less compute-intensive operations such as XORing, concatenation, and addition, as their impact is negligible compared to others. Regarding notations, we denoted the number of random operations, hashes, encryptions, and PUF operations by, R, H, E, and P, respectively. We determined the number of operations on the device side, the server side, and the total amount. In Table 5, the timings of the different operations are included in order to gain a better understanding of the comparison of the overhead. As can be seen from Table 6, our scheme is not worse considering the cost of the method in [31]. However, the cost is acceptable compared to the others, taken also into account that we offer more security features, as mentioned in Table 1.

## 8. Conclusions

Security for embedded systems and IoT platforms is gaining importance. The upcoming IoT platforms that offer integrated in-silicon security features are becoming more generalized in modern microcontroller technologies. Until very recently, security features were mostly limited to higher-end platforms running multiple cores at a core frequency of 600 MHz or higher. With the emergence of ARM Cortex-M23 and Cortex-M33, many applications involving low-end to mid-range platforms can be envisaged. We researched the feasibility of utilizing a PIC32CM5164LS00064 microcontroller based on the ARM Cortex-M23 architecture for low-power environmental monitoring. This microcontroller contains multiple security features such as TrustZone, secure boot, encryption, and hashing mechanisms. We investigated the impact of TrustZone, encryption with symmetric keys, and hashing on the development of the application. The impact was assessed in terms of code usage, RAM usage, and the required time to perform the required actions. Our results show that the TrustZone technology mostly impacts the required amount of memory, both code flash and RAM, while the performance impact on processing speed remains very limited. Encryption and hashing were evaluated on the required amount of code flash, processing time, and the additional number of bytes required during the communication between the embedded device and a remote server. The results show that cryptographic operations mostly impact the number of transmitted by doubling the communications packets. The data throughput was limited to a maximum of 50 kBps. By contrast, the required additional amount of code flash and RAM remained negligible. Aside from the TrustZone and cryptographic operations, a key agreement was also implemented so that the secret symmetric key could be refreshed at desired time intervals, mitigating the ability of an undesired third party to decrypt the transmitted messages. This key agreement was designed to take advantage of physical unclonable functions. However, due to the lack of such a mechanism, we opted to combine the true random number generator with the TrustZone to mimic the behavior of such a function. This approach can be implemented on all microcontrollers offering both the TrustZone technology and a true random number generator based on a random entropy source.

## Figures and Tables

**Figure 1 sensors-24-04720-f001:**
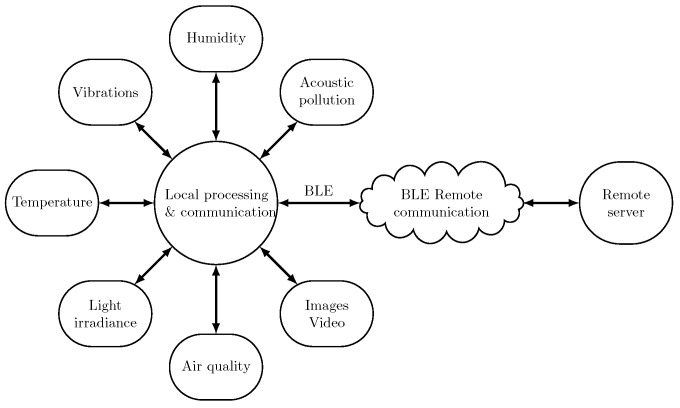
Environmental monitoring is applied by first sensing environmental variables. These variables are typically pre-processed by a microcontroller (“Local processing”) before the relevant data are transmitted to a remote server via a remote communication channel. Multiple sensing platforms can be connected via BLE to a sink platform before data are transmitted to a remote server.

**Figure 2 sensors-24-04720-f002:**
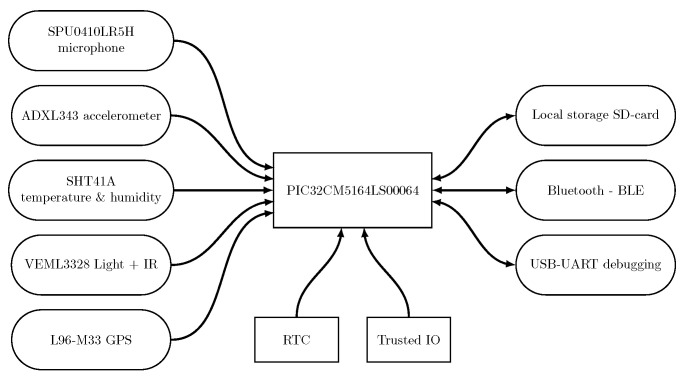
Our system enables the sensing of variables such as light and infrared (i.e., IR.) intensity, acoustic noise, temperature, humidity, and vibrations. A GPS allows us to keep track of the current location. The real-time clock (RTC) and the trusted IO are included inside the PIC32CM5164LS00064 microcontroller. Our platform allows us to store data locally on an SD card and to transmit over BLE. Debugging is provided via an onboard-included UART-to-USB converter.

**Figure 3 sensors-24-04720-f003:**
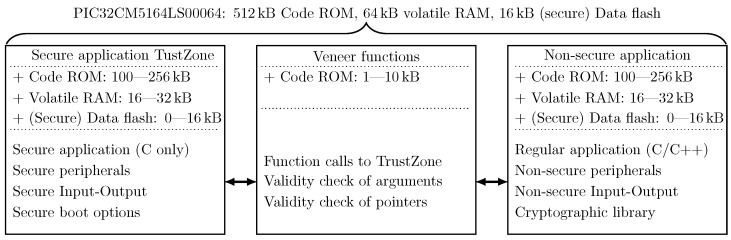
Overview of firmware implementation on PIC32CM5164LS00064 microcontroller with TrustZone enabled. The different types of memories and the peripherals need to be allocated upfront for the development of the application.

**Figure 4 sensors-24-04720-f004:**
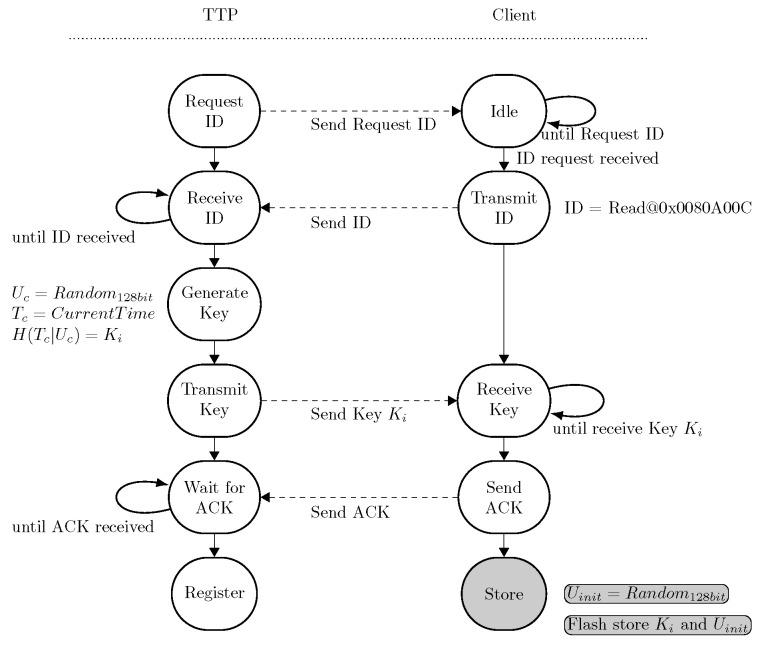
State machine of the registration phase of the client. Both the server and the client go through this state machine only once. Once this state machine is completed, both the client and the server continue to the initialization phase. The operation performed with TrustZone access (i.e., the PUF operation) is denoted with a gray state.

**Figure 5 sensors-24-04720-f005:**
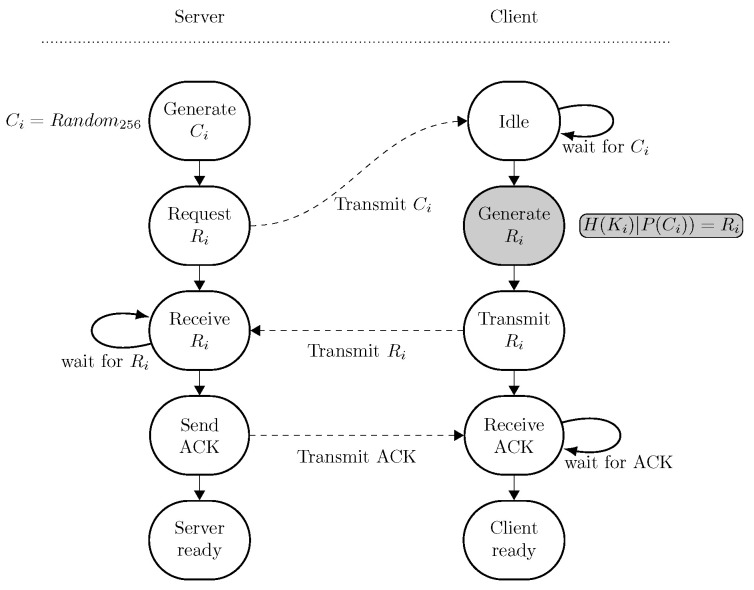
State machine of the initialization phase. Both the server and the client go through this state machine only once. Once this state machine is completed, both the client and the server continue to the “authentication and key agreement” phase. The operation performed with TrustZone access (i.e., the PUF operation) is denoted with a gray state.

**Figure 6 sensors-24-04720-f006:**
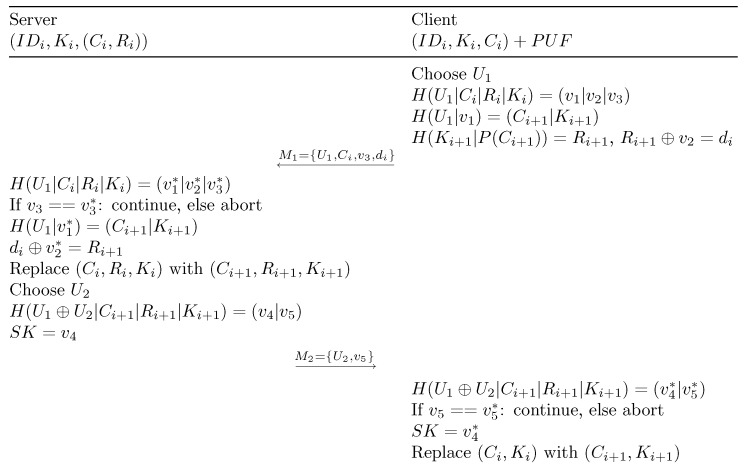
Steps and computations in the proposed authentication and key agreement protocol. Values denoted by the superscript asterisk (*) are calculated based on established data and function as internal consistency checks to verify the validity of the information received from the other party.

**Figure 7 sensors-24-04720-f007:**
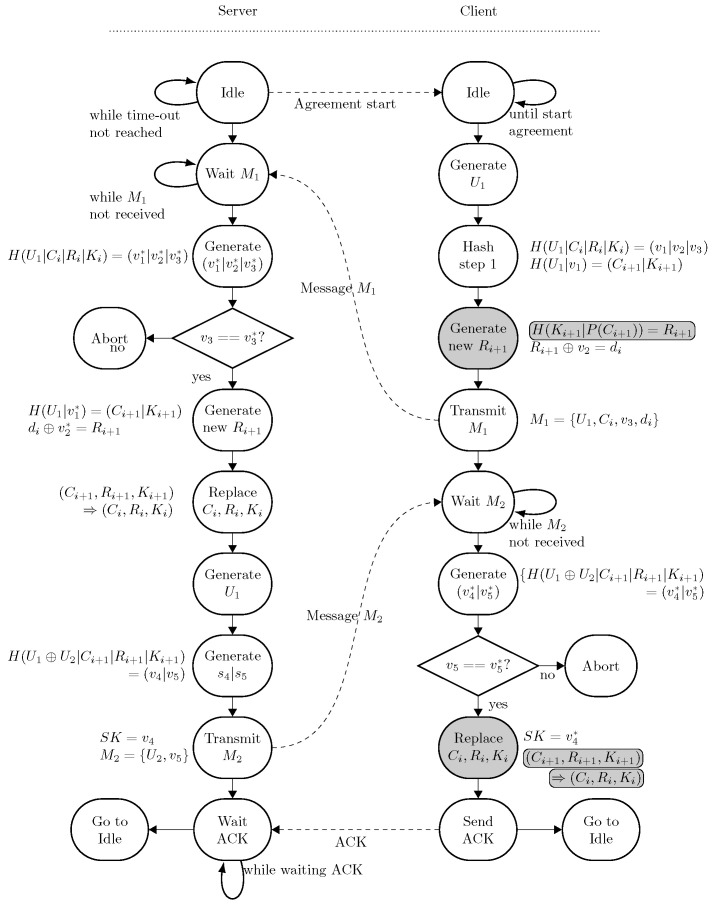
State machine of the key agreement phase of the client. Both the server and the client go through this state machine at regular definable time intervals. After a successful key agreement, both the client and the server return to their respective ‘Idle’ state. The operation performed with the TrustZone access (i.e., the PUF operation) is denoted with a gray state. Values denoted by the superscript asterisk (*) are calculated based on established data and function as internal consistency checks to verify the validity of the information received from the other party.

**Figure 8 sensors-24-04720-f008:**
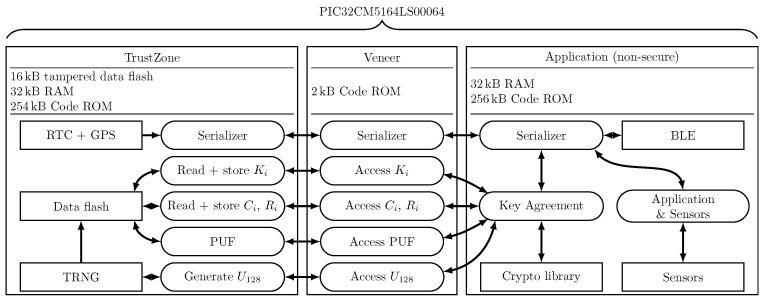
Overview of the utilized resources on the PIC32CM5164LS00064 microcontroller. The firmware functionalities and the hardware resources are encapsulated in rounded rectangles and regular-shaped rectangles, receptively.

**Figure 9 sensors-24-04720-f009:**
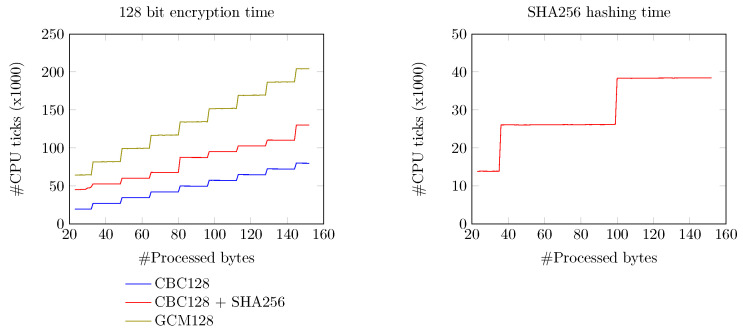
Number of CPUticks required to perform AES-128 encryption (**Left**) and SHA-256 hashing (**Right**) for a given number of input bytes. Note that the number of bytes processed represents the useful payload and excludes the header containing 20 bytes.

**Table 1 sensors-24-04720-t001:** Comparison of security features for other symmetric-key-based client–server schemes.

Scheme	Real Implementation	PUF-Based	Perfect Forward Secrecy	Anonymity/Unlinkability	Resistant to Insider Attacks	Scalable
[31]	0	0	x	x	0	x
[36]	0	0	x	x	0	x
[42]	x	x	0	0	x	0
[43]	x	x	x	0	0	0
[44]	0	x	x	0	0	0
[45]	x	x	x	0	0	0
[46]	0	x	x	0	0	x
This	x	x	x	x	x	x

**Table 2 sensors-24-04720-t002:** Number of clock CPUticks or time required to transfer a given number of bytes between the non-secure application and the secure application.

# Bytes Transferred	# Required CPUticks	Req. Transfer Time	95% Conf. Interval
16	1184	24.7 μs	±2.6 μs
32	1779	37.1 μs	±2.3 μs
48	2419	50.4 μs	±3.2 μs
64	3101	64.6 μs	±4.2 μs
80	3724	77.6 μs	±4.2 μs

**Table 3 sensors-24-04720-t003:** Usage of volatile memory (RAM) and code flash (flash) of the cryptographic methods. All units are expressed in kB.

	Used Flash	Total Flash	Used RAM	Total RAM
No crypto	207,271	261,632	15,222	32,768
With crypto	214,945	261,632	17,082	32,768
Difference	7674	-	1.86	-

**Table 4 sensors-24-04720-t004:** Comparison of communication overhead between our scheme and other related protocols in the literature. Note that the authors in [36] have 2 versions of their protocol.

Scheme (Authors)	# Phases	# Bits Sent
[31]	2	2305
Version 1 of [36]	2	896
Version 2 of [36]	2	1426
[42]	3	896
[46]	3	1280
This	2	1024

**Table 5 sensors-24-04720-t005:** Time required to perform several actions during the key agreement state machine.

Step	# Required CPUticks	Req. Processing Time	95% Conf. Interval
Generate $U_{1}\left(R\right)$	530	11 μs	±0 μs
Hash step 1 (H)	18,718	390 μs	±1.56 μs
Generate $R_{i+1}$ (P)	30,500	635 μs	±3.16 μs
Transmit $M_{1}$	942	19.8 μs	±0 μs
Compute session	40,811	850 μs	±5.51 μs
Store $K_{i+1}$	49,865	1034 μs	±8.9 μs
Store $C_{i+1}$ and $R_{i+1}$	59,180	1233 μs	±9.0 μs

**Table 6 sensors-24-04720-t006:** Comparison of computation overhead between our scheme and other related protocols in the literature. Note that the authors in [36] have 2 versions of their protocol.

Scheme	Ops Device	Time (μs)	Ops Server	Time (μs)	Total Ops	Total Time (μs)
[31]	1R + 7H	2741	1R + 12H	4691	2R + 19H	7432
Version 1 of [36]	1R + 1H	401	1R + 1H	401	2R + 2H	802
Version 2 of [36]	1R + 2H	791	1R + 2H	791	2R + 4H	1582
[42]	1R + 3H + 1P	1816	2E	4124	1R + 3H + 2E + 1P	5940
[46]	6H + 1E + 2P	5675	4H + 1E	3625	10H + 2E + 2P	9300
This	1R + 4H + 1P	2206	1R + 3H	1181	2R + 7H + 1P	3387

## Data Availability

Data are contained within the article.

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
