# Peer review of "Trustworthy Environmental Monitoring Using Hardware-Assisted Security Mechanisms"

_sensors, 2024, doi:10.3390/s24144720_

Round 1

Reviewer 1 Report

Comments and Suggestions for Authors

The manuscript is well written and organized. However, I have the following comments:

1-  The paper discusses the implementation of the proposed scheme on a microcontroller, but real-world applicability may be limited. Please discuss the implementation of the scheme on more constrained microcontrollers, which are typical in IoT environments .

2- The scheme may still be vulnerable to several types of attacks, e.g. desynchronization and DoS attacks. I recommend performing validation the countermeasures to check the practical effectiveness.

3- While the proposed scheme includes multiple security features, it does not satisfy all critical security aspects. For example, many existing PUF-based key agreement protocols, including the one discussed, do not satisfy the anonymity and unlinkability security feature. Moreover, they are often not resistant to insider attacks, meaning the server can potentially take over the role of the device.

4- Many PUF-based schemes, including the one proposed in this paper, face challenges related to scalability when parts of challenges and responses are sent in clear text. This can lead to increased vulnerability to certain types of attacks, such as machine learning attacks on PUF models. Please discuss 

Author Response

The manuscript is well-written and organized. However, I have the following comments:

  • The paper discusses the implementation of the proposed scheme on a microcontroller, but real-world applicability may be limited. Please discuss the implementation of the scheme on more constrained microcontrollers, which are typical in IoT environments.

Response: Thank you for the constructive request. There are different types of IoT applications. In the field of secure environmental monitoring (see also related work), it is common to have microcontrollers which offer more IO and digital peripheral capabilities to read-out sensor values, process these values and to transmit these values to remote server for further processing. In most cases, these devices consist of a very low-power microcontroller, which remains in a (deep) sleep state until a certain amount of time has reached or an event occurred. The microcontroller then wakes up and performs the required tasks. For this case our selected microcontroller is ideal as it is based on the new low-power ARM Cortex-M23 architecture which consumes under highest load about 22mW (powermode PL2 and excluding peripherals) and can go in (deep) sleep for the mojar part of the time. Other (older) platforms use microcontroller which typically do not offer TrustZone (older PIC models, older Texas Instruments platforms, etc.) and also offer a more limited amount of memory (SRAM, flash and code memory). As stated in the manuscript, security becomes a more important problem for IoT platforms and is especially more stringent on platforms which do not offer the TrustZone technology. Our implementation relies on new features such as the TrustZone, the TRNG, secure data flash and cryptographic operations which are typically not simultaneously available on older/low-performance devices. Therefore, porting our method to platforms not supporting all these methods poses a problem in terms of security and is out of the scope of this manuscript.

  • The scheme may still be vulnerable to several types of attacks, e.g. desynchronization and DoS attacks. I recommend performing validation the countermeasures to check the practical effectiveness.

Response: Thank you for pointing this out. We have explained in Section 5.1 from a theoretical point of view the protection of the scheme against both desynchronization and DoS attacks at the level of the security protocol. In particular, for DoS attacks, it is hard to completely guarantee protection and a multi-layered security approach, including continuous network monitoring, deployment of firewalls and intrusion detection/prevention systems, etc are required. As these mechanisms are common (in contrast to the proposed security protocol), we consider them outside the scope of this paper (similar to all other papers describing a security protocol). 

  • While the proposed scheme includes multiple security features, it does not satisfy all critical security aspects. For example, many existing PUF-based key agreement protocols, including the one discussed, do not satisfy the anonymity and unlinkability security feature. Moreover, they are often not resistant to insider attacks, meaning the server can potentially take over the role of the device.

Response: We have described in Section 5.1 the protection of the security protocol against anonymity and unlinkability. We also explained its resistance against a semi-trusted TTP (cf. insider attacks). See also Table 1 for a comparison of the security features between our protocol and others proposed in the literature, demonstrating that our protocol satisfies all of these critical security features..

  • Many PUF-based schemes, including the one proposed in this paper, face challenges related to scalability when parts of challenges and responses are sent in clear text. This can lead to increased vulnerability to certain types of attacks, such as machine learning attacks on PUF models. Please discuss 

Response: Yes indeed, we have acknowledged this issue with scalability  (see text above Table 1). Since our scheme does not leak parts of the challenges and responses in clear text, it is impossible to apply machine learning attacks to reveal the PUF structure.

Reviewer 2 Report

Comments and Suggestions for Authors

A comprehensive description of a symmetric key agreement protocol with perfect forward secrecy properties. Although its design has many common traits with other previously-published schemes, the authors test it using a modern implementation on a resource constrained embedded platform, which makes it worthy of publication.

Author Response

Comment 1: A comprehensive description of a symmetric key agreement protocol with perfect forward secrecy properties. Although its design has many common traits with other previously-published schemes, the authors test it using a modern implementation on a resource constrained embedded platform, which makes it worthy of publication.

Reply 1: Many thanks for this constructive feedback. 

Reviewer 3 Report

Comments and Suggestions for Authors

1. In Section 6. Performance analysis, the authors had better to describe the overall experimental settings, including the experimental target, experimental parameter settings, performance evaluation indexex, the organization of this Section. 

2. I cannot see the comparison between the proposed protocols or methods and the existing protocols or methods. 

3. The organization of this paper should be described explicitly, the logic of this paper can be improved. 

Author Response

  1. In Section 6. Performance analysis, the authors had better to describe the overall experimental settings, including the experimental target, experimental parameter settings, performance evaluation indexex, the organization of this Section. 

Response: We acknowledge that this is indeed needed to fully understand the analysis. Therefore, we added an additional paragraph on Experimental setting. The added text is coloured in blue.

  1. I cannot see the comparison between the proposed protocols or methods and the existing protocols or methods. 

Response: Thank you for pointing this out. Table 4 compares the proposed security protocol with other related protocols in the literature in a theoretical manner. With respect to the transmission of bits, an exact comparison can be given.
Upon your request, we now also provide, besides the total number of operations, also the corresponding processing time based on the numbers provided in Table 5.

  1. The organization of this paper should be described explicitly, the logic of this paper can be improved. 

Response: Thank you for this comment. We included an additional paragraph at the end of the Introduction section to explain the structure of the paper.

Round 2

Reviewer 1 Report

Comments and Suggestions for Authors

Authors have successfully addressed all concerns 

Reviewer 3 Report

Comments and Suggestions for Authors

The authors have already revised the manuscript, its quality is improved. I think it could be accepted as it is.